# RouteLLM: Learning to Route LLMs with Preference Data

**Isaac Ong**[*1]   **Amjad Almahairi**[*2]   **Vincent Wu**[1]   **Wei-Lin Chiang**[1]   **Tianhao Wu**[1]

**Joseph E. Gonzalez**[1]   **M Waleed Kadous**[3]   **Ion Stoica**[1,2]

[1]UC Berkeley   [2]Anyscale   [3]Canva

## Abstract

Large language models (LLMs) excel at a wide range of tasks, but choosing the right model often involves balancing performance and cost. Powerful models offer better results but are expensive, while smaller models are more cost-effective but less capable. To address this trade-off, we introduce a training framework for learning efficient router models that dynamically select between a stronger and weaker LLM during inference. Our framework leverages human preference data and employs data augmentation techniques to enhance performance. Evaluations on public benchmarks show that our approach can reduce costs by over 2 times without sacrificing response quality. Moreover, our routers exhibit strong generalization capabilities, maintaining performance even when routing between LLMs not included in training. This highlights the potential of our framework to deliver cost-effective, high-performance LLM solutions.

## 1 Introduction

Recent advances in large language models (LLMs) have demonstrated remarkable capabilities across a wide range of natural language tasks. From open-ended conversation and question answering to text summarization and code generation, LLMs have demonstrated an impressive level of fluency and understanding (Achiam et al., 2023; Bubeck et al., 2023). This rapid progress has been enabled by a combination of architectural innovations, such as the Transformer architecture (Vaswani et al., 2017), as well as scaling up data and training infrastructure (Brown et al., 2020; Radford et al., 2019).

However, not all LLMs are created equal—there exists wide variation in the sizes of different LLMs, which in turn affects the resources required to serve them. LLMs also differ in terms of the data on which they are trained, which in turn leads to variations in the strengths, weaknesses, and capabilities of different models. Broadly speaking, larger models tend to be more capable but come at a higher cost, while smaller models tend to be less capable but cheaper to serve.

This heterogeneous landscape presents a dilemma in the practical deployment of LLMs. Although routing all user queries to the largest and most capable model ensures high-quality results, it is prohibitively expensive. Conversely, routing queries to smaller models can save costs—by more than 50x (e.g., Claude-3 Haiku vs. Opus[1])—but may result in lower quality responses, as the smaller model may not handle complex queries effectively.

*LLM routing* (Ding et al., 2024; Hu et al., 2024) offers an effective solution by first processing each user query through a *router*, which then determines the most suitable LLM to handle the query. The router can direct simpler queries to smaller models and more complex ones to larger models, thereby balancing response quality with cost efficiency.

Achieving optimal LLM routing—maximizing quality within a cost constraint or minimizing cost for a target quality—is challenging. An ideal LLM router must (1) optimize response quality while invoking a single LLM per query, minimizing cost and latency as compared to multi-LLM approaches; (2) generalize to out-of-domain queries without needing separate routers for different domains; and (3) work across a broad range of LLMs without retraining, ensuring flexibility as the LLM landscape evolves.

---

[*]Equal contribution. Correspondence to `isaacong@berkeley.edu`, `anm@anyscale.com`.

[1]Per one million output tokens: Haiku ($1.25) vs. Opus ($75)

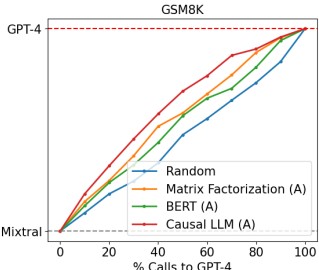 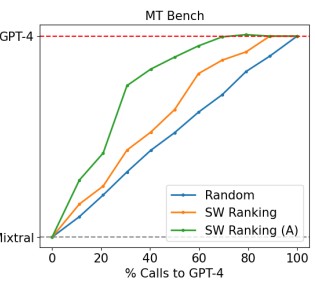 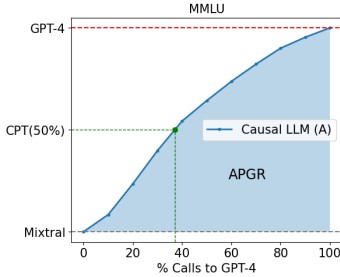

Figure 1: Routing performance/cost trade-off between GPT-4 and Mixtral-8x7B. *(left)* We demonstrate several routers that outperform the random baseline on OOD eval GSM8K. *(center)* We demonstrate improvement in router performance through data augmentation, denoted by (A), on MT Bench. *(right)* We display the main metrics we consider: call-performance threshold (CPT, denoted in green) and average performance gain recovered (APGR, denoted by the blue shaded region).

In this work, we introduce a principled framework for learning LLM routers from preference data. Our approach involves routing between two classes of models: (1) *strong models*, which provide high-quality responses at a high cost (e.g., GPT-4), and (2) *weak models*, which offer lower-quality responses at a reduced cost (e.g., Mixtral-8x7B). The objective is to minimize costs while achieving a specific performance target, e.g., 90% of the strong model, by intelligently routing simpler queries to a weak model and reserving more complex queries for the strong model. We use our framework to train several router models and evaluate them on widely recognized benchmarks such as MMLU (Hendrycks et al., 2020) and MT Bench (Zheng et al., 2023). We demonstrate that our router models significantly reduce costs—by over 2x—without substantially compromising quality. Moreover, they show strong performance across multiple strong / weak model pairs without requiring retraining.

To summarize, we make the following contributions:

- We propose a learning framework for routers that leverages human preference data and data augmentation techniques, achieving over 2x cost savings on popular benchmarks with minimal impact on response quality.

- We demonstrate that our approach enables routers to generalize to unseen data while maintaining strong performance across multiple LLMs, allowing a single trained router to be effective across a wide range of use cases.

- We open source our framework for training, serving, and evaluating LLM routers, allowing users to easily train their own routers and compare router performance across benchmarks.

## 2 RELATED WORK

A key distinction exists between reward modeling (Ouyang et al., 2022) and LLM routing. Reward modeling assesses response quality after an LLM generates it, whereas routing involves selecting the appropriate LLM beforehand. This requires a deep understanding of the query's complexity and the specific capabilities of available models.

Several recent works have also examined the cost-performance trade-offs in routing between different LLMs. LLM-Blender (Jiang et al., 2023) uses an ensemble framework that queries multiple LLMs during inference and selects the best response. Frugal-GPT (Chen et al., 2023) follows a cascading approach, sequentially querying LLMs until a reliable response is obtained. AutoMix (Aggarwal et al., 2024) uses a smaller model to self-verify its response before potentially routing the query to a larger model. These methods rely on multiple LLM queries, which can increase latency. In contrast, our approach routes each query to a single LLM, addressing the latency constraints of an ideal LLM router.

Hybrid-LLM (Ding et al., 2024) shares some similarities with our framework but differs in key aspects: it uses synthetic preference labels from the MixInstruct dataset (Jiang et al., 2023) based

on BARTScore (Yuan et al., 2021) and relies on a single BERT-based router. In contrast, we leverage human preference labels from Chatbot Arena (Chiang et al., 2024) and explore multiple router architectures, showing that data augmentation significantly boosts performance across all architectures. Additionally, Hybrid-LLM evaluates on the MixInstruct test split and lacks evidence of out-of-domain generalization, whereas we aim to demonstrate this by evaluating on several decontaminated public benchmarks.

Finally, Zooter (Lu et al., 2023) uses routing labels from QwenRM reward models (Bai et al., 2023), which can inherit biases from their training data, affecting the reliability of the routing decisions. In contrast, our approach relies mainly on human preference data. Like Hybrid-LLM, Zooter explores only a BERT-style router. Additionally, their training signal relies on a fixed set of LLMs, limiting its adaptability to other LLMs. In contrast, we show that our approach maintains strong performance even with LLMs not included in the training data.

# 3 LLM ROUTING

## 3.1 PROBLEM FORMULATION

Consider a set of LLM models $\mathcal{M}$, where each model $M : \mathcal{Q} \to \mathcal{A}$ can be viewed as a function that maps a query $q \in \mathcal{Q}$ to an answer $a = M(q) \in \mathcal{A}$. In this work, we focus on routing between two classes of models: (1) *strong models* ($\mathcal{M}_{\text{strong}}$), which are capable of producing high-quality responses but come at a high cost, such as advanced proprietary models like GPT-4 (OpenAI, 2023), and (2) *weak models* ($\mathcal{M}_{\text{weak}}$), which offer lower-quality responses but at a reduced cost, such as models like Mixtral-8x7B (Jiang et al., 2024). This binary routing problem is common in practice, especially as users seek to optimize the trade-off between quality and cost by transitioning from closed-source to open-source models. Additionally, solving the binary routing challenge provides a foundation for extending to a more complex $N$-way routing scenario.

Assume we have access to *preference data*: $\mathcal{D}_{\text{pref}} = \{(q, l_{s,w}) \mid q \in \mathcal{Q}, l_{s,w} \in L\}$, where $l_{s,w}$ represents the outcome of comparing the responses from a strong model $M_s \in \mathcal{M}_{\text{strong}}$ and a weak model $M_w \in \mathcal{M}_{\text{weak}}$ for a given query $q$, and takes values from the set $L = \{\text{win}_s, \text{tie}, \text{win}_w\}$. We introduce a principled framework for learning a binary routing function $R^\alpha : \mathcal{Q} \to \{\mathcal{M}_{\text{weak}}, \mathcal{M}_{\text{strong}}\}$ from preference data. Our approach defines $R^\alpha$ using two key components:

1) **Win Prediction Model**: This model estimates the probability that a strong model in $\mathcal{M}_{\text{strong}}$ will outperform a weak model in $\mathcal{M}_{\text{weak}}$ for a given query $q$. This probability is denoted by $P_{\boldsymbol{\theta}}(\text{win}_s|q)$, where $\boldsymbol{\theta}$ represents the model parameters. These parameters are learned by maximizing the likelihood of the observed preference data:

$$\max_{\boldsymbol{\theta}} \sum_{(q, l_{s,w}) \in \mathcal{D}_{\text{pref}}} \log P_{\boldsymbol{\theta}}(l_{s,w} \mid q). \tag{1}$$

By optimizing this likelihood, the model captures the comparative strengths and weaknesses of the two model classes across different query types. In Section 4.2, we discuss several approaches for parameterizing this win prediction model.

2) **Cost Threshold** $\alpha \in [0, 1]$: This threshold translates the predicted winning probability into a routing decision between $\mathcal{M}_{\text{weak}}$ and $\mathcal{M}_{\text{strong}}$. Given a query $q$, the routing decision is defined as:

$$R^\alpha(q) = \begin{cases} \mathcal{M}_{\text{weak}} & \text{if } P(\text{win}_s \mid q) < \alpha, \\ \mathcal{M}_{\text{strong}} & \text{if } P(\text{win}_s \mid q) \geq \alpha. \end{cases} \tag{2}$$

The threshold $\alpha$ controls the trade-off between quality and cost: a higher value of $\alpha$ enforces stricter cost constraints by favoring weak models more often, while a lower $\alpha$ biases toward higher-quality (but more expensive) strong models.

Finally, with the routing function $R^\alpha$ and two models, $M_s \in \mathcal{M}_{\text{strong}}$ and $M_w \in \mathcal{M}_{\text{weak}}$, we define a *router model* $M_{R^\alpha} : \mathcal{Q} \times \mathcal{M}_{\text{strong}} \times \mathcal{M}_{\text{weak}} \to \mathcal{A}$, which responds to a query $q$ as follows:[2]

$$M_{R^\alpha}(q, M_s, M_w) = \begin{cases} M_s(q) & \text{if } R^\alpha(q) = \mathcal{M}_{\text{strong}}, \\ M_w(q) & \text{if } R^\alpha(q) = \mathcal{M}_{\text{weak}}. \end{cases} \tag{3}$$

---

[2]For brevity, we denote this as $M_{R^\alpha}(q)$.

## 3.2 METRICS

In this section, we define evaluation metrics to capture the trade-off between cost and performance in the LLM routing problem. We begin with metrics that independently assess the cost efficiency and performance of a router model $M_{R^\alpha}$ routing between two models $M_s \in \mathcal{M}_{\text{strong}}$ and $M_w \in \mathcal{M}_{\text{weak}}$, and then introduce two compounded metrics used in our experimental evaluations.

We measure the cost efficiency of $M_{R^\alpha}$ by calculating the *percentage of calls to strong models*:

$$c(M_{R^\alpha}) = \frac{1}{|\mathcal{Q}|} \sum_{q \in \mathcal{Q}} \mathbb{I}\{R^\alpha(q) = \mathcal{M}_{\text{strong}}\}, \tag{4}$$

since $\mathcal{M}_{\text{strong}}$ models are significantly more costly than $\mathcal{M}_{\text{weak}}$ models.

For performance, we calculate the *average response quality*:

$$r(M_{R^\alpha}) = \frac{1}{|\mathcal{Q}|} \sum_{q \in \mathcal{Q}} s(M_{R^\alpha}(q)), \tag{5}$$

where $s(M_{R^\alpha}(q))$ represents an individual response quality score, such as correctness on golden-labeled datasets (e.g., MMLU) or a numerical rating (e.g., 1-5 or 1-10), with higher values indicating better quality. Similarly, $r(M_s)$ and $r(M_w)$ can be defined for the strong and weak model respectively.

Since the router model's performance falls between that of the weak and strong models, we quantify its performance relative to the gap between them. We define the router's overall performance improvement using the *performance gap recovered (PGR)*:

$$PGR(M_{R^\alpha}) = \frac{r(M_{R^\alpha}) - r(M_w)}{r(M_s) - r(M_w)}. \tag{6}$$

This metric captures how much of the performance difference between the weak and strong models is recovered by the router model.

Neither of the above metrics alone is sufficient to capture the quality-cost trade-off in routing. For example, a trivial router that always sends queries to the strong model achieves a perfect $PGR = 1$ but with no cost savings. To address this, we compute a *call-performance graph* for a router $M_{R^\alpha}$ by varying the threshold values $\alpha$. We then define the **average performance gap recovered (APGR)** as an overall measure of the router's ability to recover the performance gap under different cost constraints:

$$APGR(M_{R^\alpha}) = \int_0^1 PGR(M_{R^\alpha}) \, d\left(c(M_{R^\alpha})\right). \tag{7}$$

In Figure 1-*(right)*, APGR corresponds to the area between the router's performance curve and the weak model's performance curve. Empirically, we discretize the percentage of calls over the interval $[0\%, 100\%]$ into $\{c_i\}_{i \in [10]}$. For each $c_i$, we determine the threshold $\alpha_i$ that satisfies the cost constraint. We approximate $APGR$ as:

$$APGR(M_{R^\alpha}) \approx \frac{1}{10} \sum_{i=1}^{10} PGR(M_{R^{\alpha_i}}) \tag{8}$$

In many real-world applications, it is essential to quantify the cost required to achieve a certain level of performance. To address this, we introduce a second metric called **call-performance threshold (CPT)**. Given a desired router performance, i.e., achieving a PGR of $x\%$, the CPT$(x\%)$ represents the *minimum percentage* of calls to the strong model needed to reach the desired PGR. In Figure 1-*(right)*, the dotted green line illustrates CPT(50%), indicating the percentage of calls to GPT-4 needed to achieve a PGR of 50%. In this figure, $CPT(50\%) \approx 37\%$.

## 4 METHODOLOGY

### 4.1 CHATBOT ARENA DATA

Our primary source for preference data is the 80k battles from the online Chatbot Arena platform (Chiang et al., 2024), where users submit prompts and receive responses from two anonymous models.

After reviewing the responses, users vote for a winner or declare a tie. This generates a dataset, denoted as $\mathcal{D}_{\text{arena}}$, which contains user queries, model responses, and pairwise comparison labels based on human judgment.

A key challenge with the raw Chatbot Arena data is label sparsity. On average, the percentage of comparison labels between any two models is less than 0.1%. To address this, we derive the preference data by clustering the models in the data into 10 tiers (see Appendix A) based on their scores on the Chatbot Arena leaderboard[3], and minimize intra-tier variation using dynamic programming. We choose models in the top two tiers to represent the $\mathcal{M}_{\text{strong}}$ class, and models in the third tier represent the $\mathcal{M}_{\text{weak}}$ class. Crucially, we exclude model responses and retain only the winner identities in training. The resulting dataset is defined as $\mathcal{D}_{\text{arena}} = \{(q, l_{s,w}) \mid q \in \mathcal{Q}, l_{s,w} \in L\}$.

### 4.1.1 DATA AUGMENTATION

Despite classifying models into tiers, the human preference signal remains sparse across different model classes. As discussed in Sec 5.1, this sparsity can limit generalization, particularly for parameter-heavy router models. To address this, we explore two data augmentation methods:

**Golden-labeled datasets**: We augment our training data with labeled datasets of the form $\mathcal{D}_{\text{gold}} = \{(q, l_g, l_{s,w}) \mid q \in \mathcal{Q}, l_g \in \mathbb{R}, l_{s,w} \in L\}$, where a golden label $l_g$ is the known correct answer, e.g. in multiple-choice questions. Specifically, we use the validation split of the MMLU multiple choice benchmark (Hendrycks et al., 2020) containing approximately 1500 questions and derive comparison labels $l_{s,w}$ simply by comparing the responses from $M_s$ and $M_w$ to the golden label.

**LLM-judge-labeled datasets**: We explore obtaining preference labels on open-ended purpose chat domains using a LLM judge (Zheng et al., 2023), as it has demonstrated a high correlation with human judgment (Dubois et al., 2024; Jiang et al., 2023). Given a collection of user queries, we start by generating responses from both a strong model $M_s \in \mathcal{M}_{\text{strong}}$ and a weak model $M_w \in \mathcal{M}_{\text{weak}}$, then producing pairwise comparison labels using GPT-4 as a judge. The primary challenge with this method is the high cost of collecting responses and pairwise comparisons from GPT-4 in large quantities. Fortunately, the Nectar dataset (Zhu et al., 2023) offers a wide variety of queries with corresponding model responses. We significantly reduce our costs by selecting queries with GPT-4 responses (as $M_s$), on which we generate responses from Mixtral-8x7B (as $M_w$). Finally, we obtain pairwise comparison labels using the GPT-4 judge.[4] Overall, we collect a preference dataset $\mathcal{D}_{\text{judge}}$ of approximately 120K samples costing around $700 USD in total.

### 4.2 ROUTING APPROACHES

We now discuss several methods to define the win prediction model $P_{\boldsymbol{\theta}}(\text{win}_s | q)$ introduced in Eq 1.

**Similarity-weighted (SW) ranking** We adopt a Bradley-Terry (BT) model (Bradley & Terry, 1952) similar to Chiang et al. (2024). Given a user query $q$, we compute its cosine similarity to each query $q'$ in the train set, scaled according to the maximum cosine similarity for $q'$ in the dataset:

$$\mathcal{S}(q, q') = \frac{\epsilon \cdot \epsilon'}{\|\epsilon\| \|\epsilon'\| \cdot \max_{1 \le s \le |\mathcal{D}_{\text{pref}}|} \frac{\epsilon' \cdot \epsilon_s}{\|\epsilon'\| \|\epsilon_s\|}}, \tag{9}$$

where $\epsilon$ and $\epsilon'$ denote text embeddings for $q$ and $q'$ respectively. This similarity score is used to compute a weight scalar for each training query $\omega' = \gamma^{1+\mathcal{S}(q,q')}$.[5] We learn BT coefficients $\xi_s, \xi_w$ for the strong and weak models by solving:

$$\underset{\xi_s, \xi_w}{\operatorname{argmin}} \sum_{(q, l_{s,w}) \in \mathcal{D}_{\text{pref}}} \left[ \omega' \cdot \ell \left( l_{s,w}, \frac{1}{1 + e^{\xi_s - \xi_w}} \right) \right], \tag{10}$$

where $\ell$ is a binary cross-entropy loss. These coefficients allow us to estimate the win probability as: $P_{\boldsymbol{\theta}}(\text{win}_s | q) = \frac{1}{1 + e^{\xi_s - \xi_w}}$. In this approach, no training is required—solving is performed at inference time.

---

[3]https://leaderboard.lmsys.org

[4]We employ best practices recommended in (Zheng et al., 2023) to de-bias GPT-4 judgements

[5]We find that exponential scale works best in practice and choose $\gamma = 10$.

**Matrix factorization** Drawing inspiration from matrix factorization models used in recommendation systems to capture low-rank structures in user-item interactions (Koren et al., 2009; Töscher et al., 2009), we apply this approach to learn from preference data. The goal is to uncover a hidden scoring function $\delta : \mathcal{M} \times \mathcal{Q} \rightarrow \mathbb{R}$, where $\delta(M, q)$ represents the quality of the model $M$'s response to query $q$. If $M$ is better than $M'$ on a query $q$, then $\delta(M, q) > \delta(M', q)$. We enforce this by modeling the win probability using a sigmoid function $\sigma$:

$$P_{\boldsymbol{\theta}}(\text{win}_s|q) = \sigma\left(\delta(M, q) - \delta(M', q)\right), \tag{11}$$

which we optimize on the preference data. The scoring function $\delta$ is modelled as a bilinear function of the model and query embeddings. We embed the model $M$ into a $d_m$-dimensional vector $v_m$, and the query $q$ into a $d_q$-dimensional vector $v_q$:

$$\delta(M, q) = w_2^T(v_m \odot (W_1^T v_q + b)), \tag{12}$$

where $\odot$ denotes the Hadamard product. $W_1 \in \mathbb{R}^{d_q \times d_m}$ and $b \in \mathbb{R}^{d_m}$ are parameters of a projection layer to align the dimension of $v_q$ with $v_m$. $w_2 \in \mathbb{R}^{d_m}$ is the linear regression layer to produce the final scalar. This method is essentially learning a matrix factorization of the score matrix on the set $\mathcal{Q} \times \mathcal{M}$. We train the model on a 8GB GPU for $\approx 10$ epochs, using batch size 64 and the Adam optimizer (Kingma & Ba, 2017) with learning rate $3 \times 10^{-4}$ and weight decay $1 \times 10^{-5}$.

**BERT classifier** We explore using a standard text classification method with a higher number of parameters compared to previous methods. We use a BERT$_{\text{BASE}}$ architecture (Devlin et al., 2018), to give a contextualized embedding of the user query, and define win probability as:

$$P_{\boldsymbol{\theta}}(\text{win}_s|q) = \sigma(W h_{\text{CLS}} + b), \tag{13}$$

where $h_{\text{CLS}}$ is an embedding corresponding to the special classification token (CLS) summarizing the input query $q$. $W$ and $b$ are parameters of the logistic regression head, while $\sigma$ is the sigmoid function. We perform full-parameter fine-tuning on $\mathcal{D}_{\text{pref}}$. We train the model on 2xL4 24GB GPUs for $\sim 2000$ steps using a batch size of 16, maximum sequence length of 512, learning rate of $1 \times 10^{-5}$ and a weight decay of 0.01.

**Causal LLM classifier** We finally expand the capacity of our router by parameterizing it with Llama 3 8B (AI@Meta, 2024b). We use an instruction-following paradigm (Wei et al., 2021), i.e. we provide as input an instruction prompt containing the user query and output the win probability in a next-token prediction fashion, instead of using a separate classification head. Notably, we append the comparison labels as additional tokens to the vocabulary, and compute the win probability as a softmax over the label classes $\mathcal{L}$. We train the model on 8xA100 80GB GPUs for $\sim 2000$ steps using a batch size of 8, maximum sequence length of 2048, and a learning rate of $1 \times 10^{-6}$.

## 5 EXPERIMENTS

**Training data**: As mentioned in Sec. 4.1, we primarily use the 80K Chatbot Arena data $\mathcal{D}_{\text{arena}}$ for training our models, but hold out 5k samples for validation. We prune all prompt samples shorter than 16 characters, resulting in 65k pairwise comparisons between 64 different models. These consist of conversations from over 100 languages, with the bulk of the conversations (81%) in English, followed by Chinese (3.1%), and Russian (2.2%). We assign models to 10 classes to reduce sparsity of comparison labels. As discussed in Sec. 4.1.1, we further augment our training data with with either: 1) $\mathcal{D}_{\text{gold}}$, golden-labeled data created from the MMLU validation split, or 2) $\mathcal{D}_{\text{judge}}$, GPT-4-as-a-judge labeled data.

**Evaluation benchmarks**: We evaluate our routers on three widely-used academic benchmarks: MMLU (Hendrycks et al., 2020) consisting of 14,042 questions across 57 subjects, MT Bench (Zheng et al., 2023) with 160 open-ended questions using LLM-as-a-judge, and GSM8K (Cobbe et al., 2021) with over 1,000 grade school math problems. Additionally, we conduct a cross-contamination check between our evaluation and training datasets, and report uncontaminated results below. We present results on public benchmarks to understand the out-of-domain generalization of our routers.

**Routers**: For both the matrix factorization router and the SW ranking router, we use OpenAI's embedding model `text-embedding-3-small` to embed the input query. We perform full-parameter finetuning on both BERT and Causal LLM, and use the validation set for model selection.

We opt to use `gpt-4-1106-preview` (OpenAI, 2023) as $M_s \in \mathcal{M}_{\text{strong}}$ and Mixtral 8x7B (Jiang et al., 2024) as $M_w \in \mathcal{M}_{\text{weak}}$ to concretely evaluate router performance. We use a random router that routes queries randomly under a cost constraint as the baseline.

## 5.1 RESULTS

| Training data | Method | $CPT(50\%)$ | $CPT(80\%)$ | $APGR$ | Improvement |
|---|---|---|---|---|---|
| | Random (95% CI) | 49.03($\pm$4)% | 78.08($\pm$3)% | 0.500($\pm$0.02) | (+0%) |
| $\mathcal{D}_{\text{arena}}$ | BERT | 78.09% | 87.64% | 0.391 | (-21.8%) |
| | Causal LLM | 28.82% | 77.53% | 0.573 | (+14.6%) |
| | Matrix Factorization | **25.32%** | 74.26% | 0.580 | (+16%) |
| | SW Ranking | 37.85% | **58.99%** | **0.610** | (+22.1%) |
| $\mathcal{D}_{\text{arena}} + \mathcal{D}_{\text{judge}}$ | BERT | 19.58% | 34.02% | 0.751 | (+50.2%) |
| | Causal LLM | 31.50% | 48.75% | 0.679 | (+35.8%) |
| | Matrix Factorization | **13.40%** | **31.31%** | **0.802** | (+60.4%) |
| | SW Ranking | 23.21% | 36.04% | 0.759 | (+51.8%) |

Table 1: MT Bench results. Note that the MT Bench score at CPT(50%), 8.8, is 95% that of GPT-4's score (9.3). Our routers exhibit strong performance on MT Bench when trained on $\mathcal{D}_{\text{arena}}$, with further improvement when the dataset is augmented with $\mathcal{D}_{\text{judge}}$, reducing costs by up to 75% as compared to the random router.

Table 1 displays our router performance on MT Bench. For routers trained on the Arena dataset, we observe strong performance for both matrix factorization and similarity-weighted ranking, with both routers performing significantly better than the random router across all metrics. Notably, matrix factorization requires half the number of GPT-4 calls as compared to random to achieve a PGR of 50%. However, our BERT and causal LLM classifiers perform close to random when trained on the Arena dataset, which we attribute to high capacity approaches performing worse in a low-data regime.

Augmenting the preference data using a GPT-4 judge leads to notable improvements across all routers. The BERT and causal LLM routers now perform much better than the random baseline, with the BERT classifier achieving an APGR improvement of over 50% as compared to random. When trained on this augmented dataset, matrix factorization is the best-performing router with its CPT(80%) nearly halved and requiring 50% less GPT-4 calls as compared to random.

We also compare the MT Bench performance of our routers against existing commercial routing systems in Appendix E, demonstrating how our routers achieve substantial improvements over other available systems.

| Training data | Method | $CPT(50\%)$ | $CPT(80\%)$ | $APGR$ | Improvement |
|---|---|---|---|---|---|
| | Random (95% CI) | 50.07($\pm$0)% | 79.93($\pm$0)% | 0.500($\pm$0) | (+0%) |
| $\mathcal{D}_{\text{arena}}$ | BERT | 49.43% | 77.80% | 0.502 | (+0.5%) |
| | Causal LLM | 48.88% | 77.93% | 0.499 | (-0.2%) |
| | Matrix Factorization | **45.00%** | **76.86%** | **0.524** | (+4.9%) |
| | SW Ranking | 55.82% | 80.25% | 0.473 | (-5.4%) |
| $\mathcal{D}_{\text{arena}} + \mathcal{D}_{\text{gold}}$ | BERT | 41.30% | 72.20% | 0.572 | (+14.4%) |
| | Causal LLM | 35.49% | **70.31%** | 0.600 | (+19.9%) |
| | Matrix Factorization | 35.46% | 71.40% | 0.597 | (+19.5%) |
| | SW Ranking | **35.40%** | 71.55% | **0.603** | (+20.7%) |

Table 2: 5-shot MMLU results for our routers. Note that the MMLU score at CPT(50%), 75, is 92% that of GPT-4's score (81). Routers trained only on $\mathcal{D}_{\text{arena}}$ perform poorly due to most questions being out-of-distribution, but dataset augmentation with $\mathcal{D}_{\text{gold}}$ is highly effective, leading to significant improvement in router performance even with a small number of samples.

On MMLU (Table 2), all routers perform poorly at the level of the random router when trained only on Arena dataset, which we attribute to most MMLU questions being out-of-distribution (see Section 5.3). However, augmenting the training dataset with golden-label data from the MMLU validation split leads to significant performance improvements on MMLU across all routers, with all routers requiring approximately 20% less GPT-4 calls than random for CPT(50%). Importantly, this is despite the fact that the additional golden-labeled dataset of approximately 1500 samples represents less than 2% of the overall training data, demonstrating the effectiveness of dataset augmentation even when the number of samples is small.

| Training data | Method | $CPT(50\%)$ | $CPT(80\%)$ | $APGR$ | Improvement |
|---|---|---|---|---|---|
| | Random (95% CI) | 50.00(±2)% | 80.08(±1)% | 0.497(±0.01) | (+0%) |
| $\mathcal{D}_{\text{arena}}$ | BERT | 58.78% | 83.84% | 0.438 | (-11.8%) |
| | Causal LLM | 56.09% | 83.56% | 0.461 | (-7.3%) |
| | Matrix Factorization | **53.59%** | 85.24% | 0.4746 | (-4.5%) |
| | SW Ranking | 54.43% | **82.11%** | **0.4753** | (-4.3%) |
| $\mathcal{D}_{\text{arena}} + \mathcal{D}_{\text{judge}}$ | BERT | 44.76% | 79.09% | 0.531 | (+6.9%) |
| | Causal LLM | **33.64%** | **63.26%** | **0.622** | (+25.3%) |
| | Matrix Factorization | 38.82% | 72.62% | 0.565 | (+13.8%) |
| | SW Ranking | 41.21% | 72.20% | 0.568 | (+14.3%) |

Table 3: 8-shot GSM8K results. Note that the GSM8K score at CPT(50%), 75, is 87% that of GPT-4's score (86). Routers trained only on $\mathcal{D}_{\text{arena}}$ again perform poorly due to questions being out-of-distribution, but augmentation with $\mathcal{D}_{\text{judge}}$ substantially improves router performance.

Finally, on GSM8K (Table 3), we observe that similar to MMLU, the performance of all routers trained only on the Arena dataset is close to random. However, training our routers on the dataset augmented with synthetic data from an LLM judge substantially improves performance, with all routers going from an APGR worse than random to an APGR greater than random. When trained on this augmented dataset, the causal LLM classifier performs the best out of all routers, requiring 17% less GPT-4 calls than random to achieve CPT(50%) and CPT(80%).

## 5.2 ADAPTABILITY ACROSS MODELS

We picked `gpt-4-1106-preview` and Mixtral 8x7B as $M_s$ and $M_w$ respectively for the above experiments. However, to demonstrate the adaptability of our routers to new LLMs, we report in Table 4 the performance of our routers on MT Bench when they are used to route between two new model pairs: (1) $M_s$ = Claude 3 Opus, $M_w$ = Claude 3 Sonnet (Anthropic, 2024) and (2) $M_s$ = Llama 3.1 70B, $M_w$ = Llama 3.1 8B (AI@Meta, 2024a). Importantly, we use the same routers as before *without any retraining*, and only replace the strong and weak model routed to. These LLMs are also not present in the training data.

| Model Pair ($M_s$ / $M_w$) | Method | $CPT(50\%)$ | $CPT(80\%)$ | $APGR$ | Improvement |
|---|---|---|---|---|---|
| Claude 3 Opus / Claude 3 Sonnet | Random (95% CI) | 49.89 (±3)% | 72.27 (±4)% | 0.493 (±0.033) | (+0%) |
| | BERT | 34.85% | 39.04% | 0.682 | (+38.3%) |
| | Causal LLM | 28.12% | 50.00% | 0.656 | (+33.1%) |
| | Matrix Factorization | 31.86% | **36.43%** | 0.762 | (+54.6%) |
| | SW Ranking | **23.27%** | 51.85% | **0.772** | (+56.6%) |
| Llama 3.1 70B / Llama 3.1 8B | Random (95% CI) | 47.52(±3)% | 76.26(±2)% | 0.512 (±0.017) | (+0%) |
| | BERT | 30.15% | 38.91% | 0.673 | (+31.4%) |
| | Causal LLM | 34.05% | 45.96% | 0.689 | (+34.6%) |
| | Matrix Factorization | 25.83% | 37.30% | 0.738 | (+44.1%) |
| | SW Ranking | **21.18%** | **29.39%** | **0.767** | (+49.8%) |

Table 4: MT Bench results for our routers when used to route between different model pairs. We use the exact same routers as before trained on $\mathcal{D}_{\text{arena}} + \mathcal{D}_{\text{judge}}$. Our routers generalize very well across different model pairs without any retraining.

We observe strong results across all existing routers on MT Bench even when the model pair is replaced, with performance comparable to that of the original model pair. The results continue to be significantly stronger than random, with our best routers requiring approximately half the GPT-4 calls of the random router to achieve CPT(80%) when routing between both the Claude 3 and Llama 3.1 family of models. These results suggest that our routers have learned common characteristics of queries that allow them to distinguish between strong and weak models, generalizing to new models at inference time without additional training.

### 5.3 Quantifying dataset and benchmark similarity

We attribute the difference in the performance of routers trained on the same dataset across different benchmarks to the differing distributions of evaluation data and training data. For each benchmark-dataset pair, we compute a *benchmark-dataset similarity score* in Table 5 indicating how well-represented evaluation data is in the training data, detailed in Appendix C.

|  | Arena | Arena augmented with $\mathcal{D}_{\text{judge}}$ | Arena augmented with $\mathcal{D}_{\text{gold}}$ |
|---|---|---|---|
| MT Bench | 0.6078 | 0.6525 | - |
| MMLU | 0.4823 | - | 0.5678 |
| GSM8K | 0.4926 | 0.5335 | - |

Table 5: Benchmark-dataset similarity scores demonstrate a strong correlation between these scores and the performance of routers on the corresponding benchmarks, providing a way of quantitatively improving router performance.

A higher benchmark-dataset similarity score is correlated with stronger performance on that benchmark for routers trained using the corresponding dataset, as shown in Section 5.1. Dataset augmentation, be it using golden-labeled or LLM-judge-labeled datasets, shifts the overall distribution of the preference data to be more in line with the benchmarks and increases the benchmark-dataset similarity score, which translates into improved performance. This similarity score is also useful for understanding the relative performance of routers across different benchmarks: on $\mathcal{D}_{\text{arena}}$, the similarity score between MT Bench and all datasets is noticeably greater than other benchmarks, which we believe explains the relatively stronger router performance on MT Bench as compared to GSM8K and MMLU. Benchmark-dataset similarity scores are a promising direction for systematically improving router performance in real-world use cases, given knowledge about the query distribution.

### 5.4 Cost analysis

|  | $CPT(50\%)$ | $CPT(80\%)$ |
|---|---|---|
| MT Bench | 3.66 (95% GPT-4 quality) | 2.49 |
| MMLU | 1.41 (92% GPT-4 quality) | 1.14 |
| GSM8K | 1.49 (87% GPT-4 quality) | 1.27 |

Table 6: Cost saving ratio of our best performing routers over GPT-4. Our routers are able to achieve significant cost savings while maintaining quality.

We estimate the average cost of using GPT-4 and Mixtral 8x7B to be $24.7 per million tokens and $0.24 per million tokens respectively (detailed in Appendix D). Based on this, in Table 6, we quantify the cost savings achieved by our approach. To do so, we calculate the inverse of the ratio of GPT-4 calls made by our top-performing router relative to the random baseline because the cost of GPT-4 is the dominant factor in our analysis. The results show that our routers achieve cost savings of up to 3.66x, demonstrating that routing can significantly reduce cost while maintaining response quality.

## 5.5 ROUTING OVERHEAD

|  | Cost / million requests | Requests / second | Hourly cost of VM |
|---|---|---|---|
| SW Ranking | $39.26 | 2.9 | $0.39 |
| Matrix Factorization | $3.32 | 155.16 | $0.8 |
| BERT | $3.19 | 69.62 | $0.8 |
| Causal LLM | $5.23 | 42.46 | $0.8 |

Table 7: Cost and inference overhead of different routers. As compared to the cost of LLM generation, the cost of deploying a router is small while also able being able to support real-world workloads.

A concern with LLM routing is the overhead of routing as compared to using a single model. Therefore, we measure and report the overhead of our routers in Table 7 using randomly-sampled conversations from Chatbot Arena. For each router, we first profile the rate at which it can process requests, then use the VM's hourly cost to calculate the cost overhead. For routers that require embeddings, we also include the cost of embedding generation based on the average input length of the training set. For routers that use GPUs, namely matrix factorization and the classifier methods, we utilize Google Cloud's `g2-standard-4` VM containing a single NVIDIA L4 GPU. For similarity-weighted ranking, we use Google Cloud's CPU-only `n2-standard-8` VM.

Our GPU-based routers are currently much more efficient that our CPU-based routers, but we note that there is still much room for improvement in optimizing these routers. Based on the results, our most expensive router, SW ranking, currently adds an extra cost of no more than 0.4% as compared to GPT-4 generation (detailed in Appendix D), demonstrating the cost-effectiveness of these routers.

## 6 CONCLUSION

We demonstrate strong performance by our routers across a variety of benchmarks from open-ended question answering to humanities and math problems. By intelligently routing queries between a strong and weak model, our routers achieve significant cost savings and high response quality without excessive cost or latency overhead. We also show that our routers maintain their performance across multiple strong / weak model pairs without retraining–an important capability that if absent, would greatly limit usefulness.

Our results highlight the effectiveness of dataset augmentation in improving router performance. While training routers solely on $\mathcal{D}_{arena}$ results in poor performance on MMLU and GSM8K, augmenting the training data with an LLM judge or in-domain data enables our routers to outperform the random baseline across all benchmarks. The greatest performance gains occur when the training data closely resembles the evaluation data, as indicated by the benchmark-dataset similarity score. We believe that this framework provides a clear path towards improving routing performance for specific use cases.

While our work demonstrates strong results, there are a few limitations. First, although we evaluate on a diverse set of benchmarks, real-world applications may have distributions that differ substantially from these benchmarks. To this end, we show that users can collect a small amount of in-domain data to improve performance for their specific use cases via dataset augmentation. Next, while we focus on the two-model routing setting in this work, a promising future direction would be to extend this approach to multiple models. Finally, rather than there being a single best router for all queries, the decision of which router to use should be based holistically on latency and cost requirements, as well as the types of queries handled. In our experiments, we observe that performance between different routers trained on the same dataset can vary widely on the same benchmark without a clear explanation—we leave further investigation into this for future work.

ACKNOWLEDGMENTS AND DISCLOSURE OF FUNDING

We are grateful to Kourosh Hakhamaneshi, Goku Mohandas, Arman Zharmagambetov and Anastasiia Razdaibiedina for their valuable discussions and feedback on this work. This work is in part supported by gifts from Accenture, AMD, Anyscale, Google, IBM, Intel, Microsoft, Mohamed Bin Zayed University of Artificial Intelligence, Samsung SDS, SAP, Uber, and VMware.

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

## A    ARENA MODEL TIERS

| Tier | Models |
|------|--------|
| Tier 0 | gpt-4-0125-preview, gpt-4-1106-preview |
| Tier 1 | gpt-4-0314, gpt-4-0613, mistral-medium, claude-1, qwen1.5-72b-chat |
| Tier 2 | claude-2.0, mixtral-8x7b-instruct-v0.1, claude-2.1, gemini-pro-dev-api, gpt-3.5-turbo-0314, gpt-3.5-turbo-0613, gemini-pro, gpt-3.5-turbo-0125, claude-instant-1, yi-34b-chat, starling-lm-7b-alpha, wizardlm-70b, vicuna-33b, tulu-2-dpo-70b, nous-hermes-2-mixtral-8x7b-dpo, llama-2-70b-chat, openchat-3.5 |
| Tier 3 | llama2-70b-steerlm-chat, pplx-70b-online, dolphin-2.2.1-mistral-7b, gpt-3.5-turbo-1106, deepseek-llm-67b-chat, openhermes-2.5-mistral-7b, openchat-3.5-0106, wizardlm-13b, mistral-7b-instruct-v0.2, solar-10.7b-instruct-v1.0, zephyr-7b-beta, zephyr-7b-alpha, codellama-34b-instruct, mpt-30b-chat, llama-2-13b-chat, vicuna-13b, qwen1.5-7b-chat, pplx-7b-online, falcon-180b-chat, llama-2-7b-chat, guanaco-33b, qwen-14b-chat |
| Tier 4 | stripedhyena-nous-7b, mistral-7b-instruct, vicuna-7b, qwen1.5-4b-chat, palm-2 |
| Tier 5 | koala-13b, chatglm3-6b, gpt4all-13b-snoozy |
| Tier 6 | mpt-7b-chat, RWKV-4-Raven-14B, chatglm2-6b, alpaca-13b, oasst-pythia-12b |
| Tier 7 | fastchat-t5-3b, chatglm-6b |
| Tier 8 | dolly-v2-12b, stablelm-tuned-alpha-7b |
| Tier 9 | llama-13b |

## B    DATA CONTAMINATION

We check for cross-contamination between our evaluation dataset and the preference data used for training using embedding similarity search. Embeddings are generated for the evaluation and training data using OpenAI's `text-embedding-3-small` model. For each evaluation example, we perform a similarity search across all training data with a threshold of 0.95, returning a list of contaminated examples. We discard these evaluation examples and report results on uncontaminated scores.

## C    BENCHMARK-DATASET SIMILARITY

Let $\epsilon_B = \{b_1, b_2, \ldots, b_n\}$ be the embeddings of the prompts for a given benchmark $B$ and $\epsilon_D = \{d_1, d_2, \ldots, d_m\}$ be the embeddings of a specific preference dataset $\mathcal{D}_{\text{pref}}$, where $n$ and $m$ are the total number of evaluation and preference data samples respectively. We define the *benchmark-data similarity score* $\mathcal{S}(B, \mathcal{D}_{\text{pref}})$ for each benchmark $B$ as the average maximum similarity for each evaluation prompt across all dataset samples:

$$\mathcal{S}(B, \mathcal{D}_{\text{pref}}) = \frac{1}{n} \sum_{i=1}^{n} \max_{1 \le j \le m} \frac{b_i \cdot d_j}{\|b_i\| \|d_j\|} \tag{14}$$

We opt to use only the maximum similarity score because having a small number of samples of preference data that are very similar to the user's query is most valuable for efficient query routing, as opposed to having many samples that are less similar to the user prompt.

## D    COST CALCULATION

Since our evaluations are performed with the `gpt-4-1106` endpoint, we use its pricing ($10 per 1 million input tokens and $30 per 1 million output tokens) in our analysis. For the sake of simplicity, we assume the routers will be mostly handling short prompts in a single turn setting. We find the average input prompt in the training set to be 95 tokens long, and the average output responses to be 264 tokens long. This means the input/output tokens ratio is roughly $\frac{95}{264}$. Using these information, we estimate the average cost of using GPT-4 to be: $\frac{\left(\frac{95 \times 10}{1,000,000} + \frac{264 \times 30}{1,000,000}\right) \times 1,000,000}{95 + 264} \approx 24.7$ USD per 1

million tokens. For Mixtral 8x7B, we assume the same price for both input and output tokens, which makes the average cost $0.24 USD per 1 million tokens.

# E  INDEPENDENT BENCHMARKS

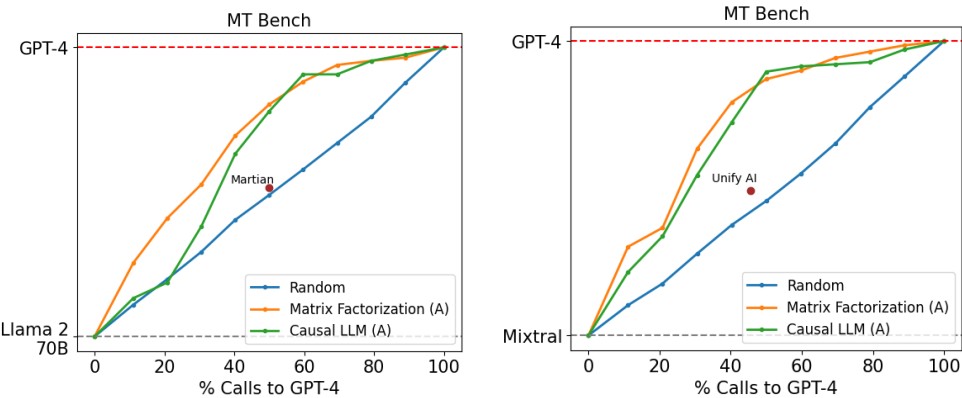

Figure 2: Performance of our routers as compared to other routing systems on MT Bench. Our routers demonstrate competitive performance, achieving stronger performance than existing routers for the same cost.

In Figure 2, we present the performance of our best-performing routers on MT Bench as compared to Unify AI (UnifyAI, 2024) and Martian (Martian, 2024), two existing commercial offerings for LLM routing.

Here, we route between `gpt-4-turbo-2024-04-09` (OpenAI, 2023) as $M_s$, and either `mixtral-8x7b-instruct-v0.1` (Jiang et al., 2024) or `llama-2-70b-chat` (Touvron et al., 2023) as $M_w$ depending on which model each system supports. For Unify AI, we select the best-performing router configuration on the user dashboard and use it for benchmarking. For Martian, we optimize for performance and specify the maximum cost per million tokens as $10.45, approximating this value using public inference costs (OpenAI, 2024; Together.AI, 2024) based on a 1:1 input:output token ratio so that 50% of calls are routed to GPT-4.

Both the matrix factorization router and causal LLM routers perform very competitively when trained on $\mathcal{D}_{\text{arena}} + \mathcal{D}_{\text{judge}}$, outperforming the commercial routing systems by achieving the same performance with up to 40% fewer calls routed to GPT-4.

# F  ADDITIONAL PLOTS

We include additional plots for the results presented in Section 5.1.

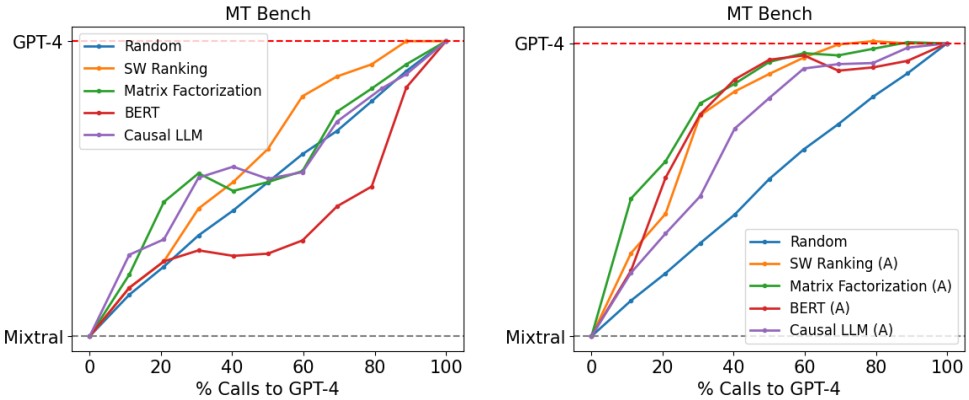

Figure 3: MT Bench performance for all routers.

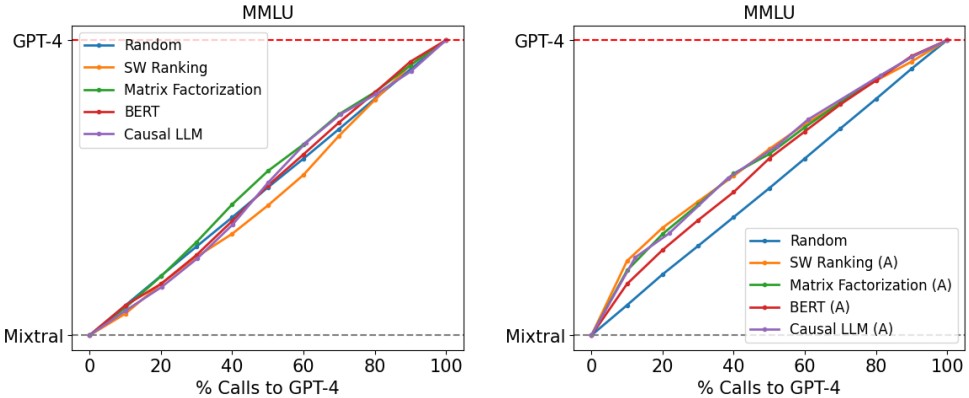

Figure 4: 5-shot MMLU performance for all routers.

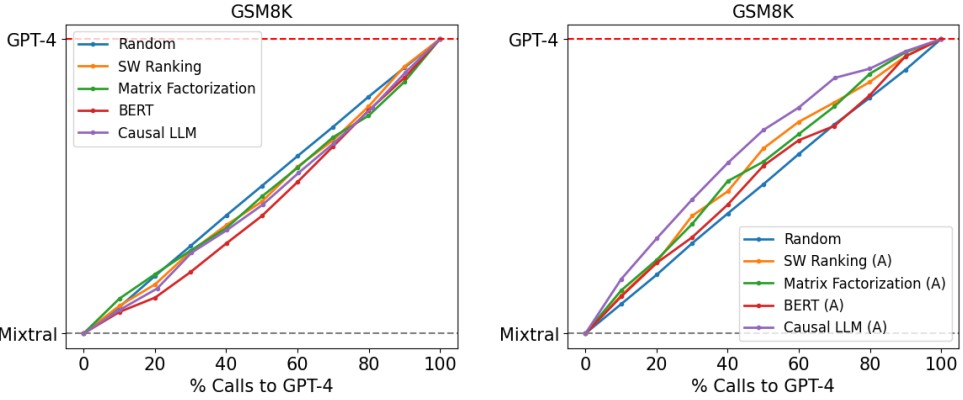

Figure 5: 8-shot GSM8K performance for all routers.

