# OpenReview forum: "RouteLLM: Learning to Route LLMs from Preference Data"
_ICLR.cc/2025/Conference — ICLR 2025 Poster_

### Official Review · Reviewer_7nQc · 2024-11-03

**Soundness:** 3
**Presentation:** 3
**Contribution:** 2
**Rating:** 5
**Confidence:** 3

**Summary:**

The paper presents a framework for training router models using human preference data and data augmentation, achieving over 2x cost savings on popular benchmarks with minimal impact on response quality. The authors employ a binary routing approach to direct simple queries to a cost-effective model (e.g., Mixtral-8x7B) and complex queries to a stronger model (e.g., GPT-4). They demonstrate generalization across unseen data and new LLMs without retraining, providing a single trained router adaptable to multiple use cases.

**Strengths:**

- Demonstrates significant cost savings (over 2x) without compromising response quality, verified on MMLU, MT Bench, and GSM8K.
- Introduces APGR to quantify performance relative to cost, effectively balancing quality and cost constraints (Eq. 7).
- Implements diverse methods for defining the win prediction model.
- Demonstrates robustness and adaptability, as the router generalizes effectively to unseen LLM pairs and domains without retraining.

**Weaknesses:**

- Lacks detailed analysis of routing patterns under different $\alpha$ values, such as which query types tend to be routed to strong vs. weak models, making it unclear how to set optimal $\alpha$ values for specific use cases (Sec. 5.1).
- Insufficient exploration of the router's decision-making robustness, especially regarding handling ambiguous queries where strong and weak models may perform similarly.
- Performance still heavily depends on data augmentation with high-cost LLM-judged labels.

**Questions:**

- Does the paper provide guidance on selecting the most suitable win prediction method across various scenarios?
- Could insights be provided on optimal $\alpha$ values for different query types, including a breakdown of routing decisions under varying $\alpha$ thresholds?

---

> ### Author Response · Authors · 2024-11-24
> **R3 rebuttal-1**
>
> Thank you for the thoughtful review. To address your points:
>
> **W1**
>
> > Lacks detailed analysis of routing patterns under different $\alpha$ values, such as which query types tend to be routed to strong vs. weak models, making it unclear how to set optimal
>  values for specific use cases
>
> We conduct an additional analysis where for each MMLU domain, we record the average predicted probability by a router that the strong model outperforms the weak model for queries in that domain. Below, we present the three domains with the highest and lowest mean predicted probability, focusing on the causal LLM, BERT, and matrix factorization routers.
>
> __Domains with highest mean predicted probability__ (in order from highest to lowest)
>
> - BERT: college mathematics, elementary mathematics, high school mathematics
> - Causal LLM: high school mathematics, college mathematics, abstract algebra
> - Matrix Factorization: elementary mathematics, high school mathematics, college chemistry
>
> __Domains with lowest mean predicted probability__ (in order from lowest to highest)
>
> - BERT: marketing, management, professional medicine
> - Causal LLM: management, marketing, public relations
> - Matrix Factorization: security studies, high school US history, sociology
>
> We observe a clear pattern that STEM-related subjects, especially mathematics, tend to be routed to the strong model while arts subjects like sociology are less likely to be routed to the strong model. This demonstrates that our routers have learned common patterns about which query types should be routed to either model.
>
> However, we note that the above results do not imply that a domain classifier would make a good router. Even within domains that are generally more suited to strong models, there exists a distribution of difficulties. For example, even though the matrix factorization router predicts elementary mathematics to be most difficult and security studies to be easiest, **6.3%** of elementary mathematics queries have a lower predicted probability than the average security studies query. Thus, users should not rely on domain classification for routing, but rather a threshold-based approach like we propose. To determine the right cost threshold, users should calibrate it based on both the target data distribution and the router used. We refer to the main rebuttal where we detail the process of calibration.
>
> **W2**
>
> > Insufficient exploration of the router's decision-making robustness, especially regarding handling ambiguous queries where strong and weak models may perform similarly.
>
> To shed more light on routers’ decision-making, we focus on the causal LLM router for our analysis here and consider its predicted probability that the strong model outperforms the weak model for all queries on the MMLU benchmark. We define ambiguous queries as ones where both the strong and weak model either get the answer correct or get the answer wrong. We find that for ambiguous queries, the average predicted probability by the router is 0.34 std devs **lower** as compared to the predicted probability for the entire dataset. This trend holds across other routers as well. This aligns with what users should expect from an ideal router because for queries where both models perform similarly, we can save costs by routing to the weaker model.
>
> We also extend this experiment to look at hard queries, which we define as queries where the strong model answers correctly but the weak model answers wrongly. Here, we find that the causal LLM router predicts the strong model to win at hard queries with an average probability that is 0.28 std devs **higher** than the entire dataset. This again aligns with what users expect from an ideal router, as difficult queries that only the strong model can answer should be routed to the strong model.
>
> **W3**
>
> > Performance still heavily depends on data augmentation with high-cost LLM-judged labels.
>
> Using the LLM judge to generate augment human preference data is one of two methods that we discuss in Section 4.1.1 for data augmentation. We believe that using in-domain data is equally effective, and we show that it is able to improve MMLU performance with only 1500 additional samples (Section 5.1), demonstrating its effectiveness at low cost. We believe that these two approaches to data augmentation provide a wide range of options for users to improve routing performance at reasonable costs.

---

> ### Author Response · Authors · 2024-11-24
> **R3 questions-1**
>
> > Does the paper provide guidance on selecting the most suitable win prediction method…?
>
> We refer the reviewer to our discussion in the main rebuttal.
>
> > Could insights be provided on optimal values for different query types, including a breakdown of routing decisions under varying thresholds?
>
> We calculate the percentage of queries across two MMLU domains, marketing and college mathematics, that get routed to the strong model by the matrix factorization router across different thresholds. We select cost thresholds based on the router’s predicted probabilities for the full MMLU, selecting the 20%, 50%, and 80% percentile probabilities as thresholds. These correspond to 0.179, 0.242, and 0.314 respectively.
>
> - α=0.179: 49.1% of *marketing* queries and 100% of *college mathematics* queries routed to strong model
> - α=0.242: 12.8% of *marketing* queries and 98% of *college mathematics* queries routed to strong model
> - α=0.314: 2.13% of *marketing* queries and 75% of *college mathematics* queries routed to strong model
>
> We clearly see that differences in routing emerge for different thresholds. With the lowest threshold, all college mathematics queries are routed to the strong model while only 49% of marketing queries are routed there. As the cost threshold increases, the number of queries routed to the strong model decreases across both domains. But, the number of marketing queries decreases significantly as compared to college mathematics, which only drops to 75%. This aligns with the idea that college mathematics queries are likelier to require the strong model as compared to marketing queries.
>
> We hope we have addressed your concerns and that you consider adjusting your score if so.

---

> ### Author Response · Authors · 2024-12-02
> **R3 Follow-up**
>
> Thank you once again for your thoughtful review. We believe we have thoroughly addressed your concerns in our rebuttal and would greatly appreciate the opportunity to engage further during the remaining rebuttal period. Please don't hesitate to share any additional questions or feedback!

---

> > ### Comment · Area_Chair_YxWN · 2024-12-02
> > **Reminder to Reviewer 7nQc from Area Chair**
> >
> > Dear Reviewer
> > Would you like to engage with the authors on their rebuttal? Please let us know if you have further comments or if the responses address your concerns?
> >
> > Thank you!

---

### Official Review · Reviewer_GuJg · 2024-11-04

**Soundness:** 3
**Presentation:** 3
**Contribution:** 2
**Rating:** 6
**Confidence:** 4

**Summary:**

Large language models (LLMs) excel at a wide range of tasks, but choosing the right model often involves balancing performance and cost. This paper proposes a routing approach, RouteLLM, to dynamically select between a stronger and weaker LLM during inference. Experiments on three real-world benchmarks demonstrate the effectiveness of RouteLLM.

**Strengths:**

S1. The proposed approach, RouteLLM, is able to achieve over 2x cost savings on popular benchmarks with
minimal impact on response quality.

S2. Authors demonstrate that RouteLLM enables routers to generalize to unseen data while maintaining strong performance across multiple LLMs.

**Weaknesses:**

W1. Unclear novelty and limited technical contribution. Training a router to harness the respective strengths of different LLMs has been widely studied [1,2,3,4]. Specifically, how to generalize LLM routing to OOD data has been studied in [5], how to use LLMs to generate more training data to help improve routing performance has been explored in [6], which authors did not compare to. Moreover, some technology (SW ranking) proposed in this paper shares unignorable similarity to prior work [7].

W2. Weak baselines. Provided the rich literature on this topic as aforementioned, considering a random router as the only baseline is insufficient in this work. The effectiveness of RouteLLM could be further demonstrated if authors could compare it to more advanced baselines (e.g., a subset from [1-6]).

W3. In Sec 5.5, authors provided the overhead analysis. Notably, SW ranking is both expensive ($37 / 1M requests) and slow (2.9 requests / second), which makes it hard to use in practice.


References:
[1] Routing to the Expert: Efficient Reward-guided Ensemble of Large Language Models, https://arxiv.org/pdf/2311.08692
[2] Hybrid LLM: Cost-Efficient and Quality-Aware Query Routing, https://arxiv.org/abs/2404.14618
[3] Fly-Swat or Cannon? Cost-Effective Language Model Choice via Meta-Modeling, https://arxiv.org/pdf/2308.06077
[4] ROUTERBENCH: A Benchmark for Multi-LLM Routing System, https://arxiv.org/pdf/2403.12031
[5] Large Language Model Routing with Benchmark Datasets, https://arxiv.org/pdf/2309.15789
[6] Routoo: Learning to Route to Large Language Models Effectively, https://arxiv.org/abs/2401.13979
[7] Chatbot Arena: An Open Platform for Evaluating LLMs by Human Preference, https://arxiv.org/abs/2403.04132

**Questions:**

Q1. In Sec 3.1, authors introduced the cost threshold \lambda. It is unclear that how to choose the right cost threshold when user have specific cost budgets.

Q2. Some details in overhead analysis are unclear. Both SW ranking and matrix factorization rely on embeddings generated by text-embedding-3-small. Are the costs incurred by embedding extraction included in the overhead analysis? Also, given that the model size ratio between BERT-base (110M) and causal LLM (8B) is 110M / 8B ~= 1%, it is surprising to see the cost overhead of BERT-base is ~60% of the causal LLM, and the achieved throughput is only 60% higher, according to Table 7. More details on how the overheads are estimated could be very helpful.

---

> ### Author Response · Authors · 2024-11-24
> **R2 rebuttal-1**
>
> Thank you for your thoughtful feedback. We will address the concerns that you have raised:
>
> **W1**
> > Unclear novelty and limited technical contribution
>
> We respectfully disagree with the assessment regarding the novelty of our work and believe it makes significant contributions beyond the referenced works. As outlined in Section 1, deploying an LLM router practically requires satisfying several criteria, and we highlight below some limitations of previous approaches with respect to it:
> - **Lack of out-of-domain generalization** [2], [4], and [5] evaluate their approaches on a held-out portion of the same training dataset. The same applies in [3], which uses all tasks except one for evaluation. Moreover, [3] is restricted to tasks with well-defined answers rather than open-domain chat data. The training data for [6] is limited to QA benchmarks such as ARC, and their evaluation is restricted to MMLU.
> - **Not flexible across different LLMs** [1] is constrained to routing among a fixed set of six LLaMA-based models and relies exclusively on preference rankings generated by the QwenRM reward model.
> - **Limited exploration of architectures** [1], [2], and [3] only explore a BERT-based router architecture, while [4] explores KNN and MLP-based routers. [5] trains a separate KNN-based “correctness predictor” for each LLM.
>
> In contrast, our work addresses the requirements of an ideal router by demonstrating effective generalization to routing across diverse models, including LLMs not seen during training (Section 5.2). Additionally, we explore a broader range of model architectures and go beyond existing efforts by open-sourcing a comprehensive framework for training, evaluating, and deploying LLM routers.
>
> As for [7], we acknowledge that the SW Ranking router is inspired by the ELO calculation from Chatbot Arena. However, our extension of their ELO algorithm to incorporate the similarity of responses is novel. Additionally, the use of a Bradley-Terry model is not unique to Chatbot Arena.
>
> **W2**
> > Weak baselines.
>
> We emphasize that the primary contribution of our work is not proposing a single model to "solve" the routing problem but introducing a comprehensive framework for training and evaluation. This addresses a significant gap, as prior works lack standardized training and evaluation methodologies. Our framework provides a foundation for future research to build upon.
> Many referenced baselines, such as [1], [2], and [3], rely on a BERT-based router, which we include among the architectures studied. However, direct comparisons with certain prior works (e.g., Hybrid-LLM) were challenging due to unavailable code and differing evaluation methodologies. Unlike prior works that focus on held-out splits from the same distribution, we prioritize out-of-domain generalization, a more practical and challenging criterion.
>
> To further validate the real-world effectiveness of our routers we conducted additional experiments:
>
> - **Evaluation against commercial solutions (Appendix E)**: On MT Bench, our best-performing routers matched the performance of commercial solutions (Unify AI and Martian) while being 40% more cost-efficient.
> - **Real-world online evaluation**: Through a collaboration with the Chatbot Arena team, we deployed our Causal LLM router on the Arena platform for live evaluation. The router achieved a 12% ELO improvement over a random baseline, demonstrating its practical effectiveness.
>
> **W3**
>
> > SW ranking is both expensive and slow, which makes it hard to use in practice.
>
> We note that this is an unoptimized version of SW Ranking - the primary reason that it’s slower than other approaches is because it is CPU-based rather than GPU-based. Therefore, building a GPU-accelerated version will lead to a noticeable improvement in performance. Our other methods offer a much better balance of performance and efficiency.
>
> Additionally, despite SW Ranking costing $37 / 1M requests, our calculations in Appendix D show that using the router ends up being 0.4% of GPT-4 generation cost, which is small as compared to the potential cost savings of routing.

---

> ### Author Response · Authors · 2024-11-24
> **R2 questions-1**
>
> > It is unclear that how to choose the right cost threshold when user have specific cost budgets.
>
> We refer the reviewer to the main rebuttal where we detail the process of calibration.
>
> > Are the costs incurred by embedding extraction included in the overhead analysis?
>
> For the cost analysis (Section 5.4), the cost of embeddings is not included because it is 100 times cheaper than the estimated cost of GPT-4 and we consider them negligible. We only consider the ratio of GPT-4 calls made by the best performing router to the random router to calculate the cost savings.
>
> For the overhead analysis (Section 5.5), we first profile the performance of the router on the specified cloud VMs to determine the number of requests that they are able to support per second (including embedding generation). Next, based on the hourly cost of the VM, we use that to calculate the cost per million requests. Therefore, the cost of embedding is not included.
>
> We will clarify these calculations in an updated version.

---

> > ### Comment · Reviewer_GuJg · 2024-12-02
> >
> > I want to thank the authors for the thoughtful responses which addressed most of my previous questions.
> >
> > > For the cost analysis (Section 5.4), the cost of embeddings is not included because it is 100 times cheaper than the estimated cost of GPT-4 and we consider them negligible.
> >
> > One important perspective of overhead analysis is to understand the overhead difference between different approaches. Since not all approaches leverage embedding extraction (e.g., BERT and Causal LLM), I feel a comprehensive overhead calculation including embedding costs is still needed.
> >
> > Also, one of my previous comments remains untouched in current response,
> >
> > > Also, given that the model size ratio between BERT-base (110M) and causal LLM (8B) is 110M / 8B ~= 1%, it is surprising to see the cost overhead of BERT-base is ~60% of the causal LLM, and the achieved throughput is only 60% higher, according to Table 7.
> >
> > I am willing to consider increasing my rating if all my questions were addressed.

---

> ### Author Response · Authors · 2024-12-02
> **R2-questions-2**
>
> We thank the reviewer for their response.
>
> We fully agree with the reviewer that it is important to incorporate the cost of embeddings in the overhead analysis to have a fair comparison between different routing approaches. Therefore, we have updated our overhead analysis from Section 5.5 such that the cost per million requests for each router now includes:
> 1) the cost of the virtual machine, and
> 2) the embedding cost for routers than leverage embeddings, namely the matrix factorization and SW ranking routers.
>
> To do so, we use the API cost of $0.020 / million tokens for `text-embedding-3-small` \[1\] and assume an average input token length of 95 tokens per request (Appendix D). We present the updated Table 7 below:
>
> |                      | Cost / million requests | Requests / second | Hourly cost of VM |
> |----------------------|------------------------|------------------|-------------------|
> | SW Ranking           | $39.26                 | 2.9              | $0.39             |
> | Matrix Factorization | $3.32                  | 155.16           | $0.8              |
> | BERT                 | $3.19                  | 69.62            | $0.8              |
> | Causal LLM           | $5.23                  | 42.46            | $0.8              |
>
> Regarding the reviewer’s second point on the differences between the requests / second for the BERT and causal LLM routers, we believe there are a few reasons:
>
> - The specific model that the BERT router uses is XLM-RoBERTa-base \[2\], which contains 279M parameters in FP32. On the other hand, the causal LLM router uses the Llama 3 8B model \[3\] which contains 8B parameters in BF16. This means the model size ratio in terms of parameters is closer to 3.5% rather than 1%.
> - Because of the different precisions of both routers, the effective FLOPs of the L4 GPU is 4 times less for the BERT router as compared to the causal LLM router: 30.3 TFLOPs for FP32 vs 121 TFLOPs for BF16 \[4\].
> - The different precisions also leads to longer time taken to transfer requests to the GPU for the BERT router as compared to the causal LLM router.
> - Moreover, we perform this benchmarking with batch size 1 to simulate an online setting, meaning that the routers are not fully FLOPs bound and data movement costs are a significant portion of the overall time. Therefore, this hurts the performance of the BERT router disproportionately as compared to the causal LLM router.
>
> We believe that these reasons all contribute to the measured performance of the BERT router being worse than expected as compared to the causal LLM router. That said, the reviewer makes an excellent point and we will ensure that this is clarified in the updated version of the paper.
>
> We hope this addresses the reviewer’s concerns and we are happy to answer any further questions.
>
> \[1\]: https://openai.com/api/pricing/
> \[2\]: https://arxiv.org/abs/1911.02116
> \[3\]: https://huggingface.co/meta-llama/Meta-Llama-3-8B
> \[4\]: https://www.nvidia.com/en-us/data-center/l4/

---

> > ### Comment · Reviewer_GuJg · 2024-12-03
> >
> > Thank you for all the efforts in addressing my comments and revising the manuscript! I have adjusted my score accordingly.

---

### Official Review · Reviewer_V25N · 2024-11-04

**Soundness:** 4
**Presentation:** 4
**Contribution:** 3
**Rating:** 8
**Confidence:** 4

**Summary:**

This paper introduces RouteLLM, a framework for training router models that direct queries between stronger and weaker LLMs to optimise cost-performance tradeoffs. It employs preference data from Chatbot Arena. Empirical results show that data augmentation is important to letting the routers outperform a random baseline on MMLU and GSM8K. It also demonstrates that the routers generalise across domains, and do not need to be retrained.

**Strengths:**

1. The paper presents a novel framework of directly using human preference data, as opposed to reward models, to route between a pair of LLMs.
2. The empirical evaluation is comprehensive, touching multiple architectures, LLMs and common benchmarks.
3. Both metrics of "average performance gap recovered" and "call-performance threshold" are clear reflections of real world considerations: the general ability for the router to close the performance gap between the better and worse model, as well as the cost to doing so for a minimum quality bar.
4. The increasing variation in LLM quality and cost makes it more important to be able to efficiently tradeoff between cost and performance. On MT Bench, RouteLLM is able to achieve comparable performance to GPT-4 with a cost saving of ~3.7x.

**Weaknesses:**

1. The paper focuses on a binary routing of "strong" versus "weak" model, but doesn't consider other binary differences between models.
2. The paper does not discuss in depth why certain architectures perform better or worse, especially with respect to the improvement in the causal LLM's performance when faced with data augmentation.

**Questions:**

1. How well do the routers do at separating models which are much closer in ability e.g. two different 7B models?
2. How well do the routers do if the two models are not strictly "stronger" or "weaker", but rather have been finetuned to do different tasks?

---

> ### Author Response · Authors · 2024-11-24
> **R1 rebuttal**
>
> Thank you for your review and comments, we are glad that you enjoyed the paper. Addressing your points:
>
> **W1**
>
> > The paper focuses on a binary routing of "strong" versus "weak" model, but doesn't consider other binary differences between models.
>
> Thank you for the suggestion. Our approach addresses a practical need by balancing cost and performance across general chat data without focusing on a specific domain or capability. We demonstrate that it generalizes across a class of models, rather than being limited to two specific ones (Section 5.2). We believe that this approach can readily extend to other binary distinctions, such as coding-specific versus generalist models or English-only versus multilingual models.
>
> **W2**
>
> > The paper does not discuss in depth why certain architectures perform better or worse, especially with respect to the improvement in the causal LLM's performance when faced with data augmentation.
>
> We refer the reviewer to the main rebuttal, where we provide discussion of different architectures and how to select them.
>
> **Q1**
>
> > How well do the routers do at separating models which are much closer in ability e.g. two different 7B models?
>
> We’ve previously experimented with using our trained routers to route between models of similar abilities and found that they perform worse during evaluations because our routers are trained specifically to exploit the difference in abilities between two models to exploit the tradeoff between cost and performance. However, we agree that training routers for models that are equal in ability but have other differences (such as domain expertise) is an exciting next direction.
>
> **Q2**
>
> > How well do the routers do if the two models are not strictly "stronger" or "weaker", but rather have been fine tuned to do different tasks?
>
> As discussed, we focus on strong and weak model pairs to address the need for balancing cost and performance, but we believe that our approach can also be extended to other binary distinctions, such as for models with task-specific strengths. This is a natural and practical extension of our work.

---

> > ### Comment · Reviewer_V25N · 2024-11-27
> >
> > Thanks for the thorough response! Most of my main concerns have been addressed!

---

### Author Response · Authors · 2024-11-24
**Main Rebuttal**

We thank all the reviewers for their insightful comments and questions! Since submission, we have conducted additional experiments in response to reviewers’ questions and sought to clarify concerns raised.

**On deciding the cost threshold** Reviewer 2 and Reviewer 3 raised questions around how users should determine what cost threshold to use for each router. Our suggested approach for users is to calibrate cost thresholds based on using a sample of the types of queries they expect to receive. Using the cost of the 2 models used for routing and their specified cost budget, users can first determine the percentage of queries that they would like to route to the stronger model. Given the sample of queries, users can then execute the router over these queries to obtain the estimated win probability for each query. Based on this, users can calculate the appropriate cost threshold to maximize routing performance while respecting the desired cost budget.

We provide a script `calibrate_threshold.py` in our open-source framework that automates this calibration (included as supplementary material in the submission). Notably, we used this process to deploy the causal LLM router for a live evaluation in collaboration with the Chatbot Arena team. Specifically, we calibrated the cost threshold based on a public dataset of 55k Chatbot Arena queries. By doing this, our router achieved a **12% ELO improvement** over the random router on new unseen queries, demonstrating its effectiveness.

**On the differences between different architectures and how to select them** Reviewer 1 and Reviewer 3 requested details about the differences between routers and guidance around router selection. To this end, we note that the selection of the optimal router requires a comprehensive evaluation of latency, cost constraints, and the availability of training data for the router model. Our experiments show that even with Chatbot Arena data, consisting of 65K human-labeled samples, surpassing the random baseline remains challenging, highlighting the complexity of the routing problem. Non-parametric methods, such as SW Ranking and MF, perform consistently well on Chatbot Arena data, often outperforming LLM-based classifiers and exhibiting stronger generalization across different benchmarks. The availability of high-quality labeled data, generated cost-effectively via the LLM-as-a-judge approach, has proven to be more critical to the success of the routing model than the specific architecture employed. Furthermore, the Causal LLM router, with its large number of parameters, demonstrates a clear reliance on this additional data to achieve competitive performance, as it is particularly susceptible to overfitting in low-data regimes. We will include this discussion in the next revision of the paper.

We are happy to answer any further questions reviewers may have.

---

### Meta-Review · Area_Chair_YxWN · 2024-12-19

**Metareview:**

The paper proposes RouteLLM to dynamically choose between a stronger and weaker LLM during inference, achieving 2x cost savings on 3 benchmarks with minimal impact on response quality. Empirical results show that data augmentation is important to letting the routers outperform a random baseline and generalise across domains without needing to be retrained.


Strengths:
* Demonstrates significant cost savings (over 2x) without compromising response quality [Reviewer V25N, 7nQc, GuJg]
* Introduces APGR to quantify performance relative to cost, effectively balancing quality and cost constraints [Reviewer 7nQc and V25N]
* Demonstrates robustness and adaptability, as the router generalizes effectively to unseen LLM pairs and domains without retraining [all reviewers]
* Authors also provide a process of calibration for estimating the right cost threshold to calibrate based on both the target data distribution and router used.

Weakness
* The model only considers  binary routing of strong and weak models, but does not consider other binary differences between models such as coding-specific versus generalist models or English-only versus multilingual models. To reviewer V25N comment on this, authors claim that it is applicable to these other binary differences between models, without any substantiations.

**Additional Comments On Reviewer Discussion:**

Authors have sufficiently engaged with reviewers during the rebuttal phase and Reviewers V25N and GuJg have agreed that their concerns have been addressed. Although Reviewer 7nQc has not engaged with the authors despite nudge from me, his comments also seem positive.

---

### Decision · Program_Chairs · 2025-01-22

Accept (Poster)